# Extraction of Corn Bract Cellulose by the Ammonia-Coordinated Bio-Enzymatic Method

**DOI:** 10.3390/polym15010206

**Published:** 2022-12-31

**Authors:** Xushuo Yuan, Jiaxin Zhao, Xiaoxiao Wu, Wentao Yao, Haiyang Guo, Decai Ji, Qingkai Yu, Liwen Luo, Xiaoping Li, Lianpeng Zhang

**Affiliations:** 1Yunnan Provincial Key Laboratory of Wood Adhesives and Glued Products, Southwest Forestry University, Kunming 650224, China; 2Jiaxing Key Laboratory of Molecular Recognition and Sensing, College of Biological, Chemical Sciences and Engineering, Jiaxing University, Jiaxing 314001, China

**Keywords:** cellulose, bio-enzymes, corn bract, ammonia, pretreatment, ligninase, hemicellulase, pectinase

## Abstract

This study explored a green and efficient method for cellulose extraction from corn bract. The cellulose extraction by the CHB (CH_3_COOH/H_2_O_2_/Bio-enzyme) method and the N-CHB (NH_3_·H_2_O-CH_3_COOH/H_2_O_2_/Bio-enzyme) method were compared and analyzed. The effect of ammonia pretreatment on cellulose extraction by bio-enzymatic methods was discussed. The results showed that ammonia promoted the subsequent bio-enzymatic reaction and had a positive effect on the extraction of cellulose. Sample microstructure images (SEM) showed that the cellulose extracted by this method was in the form of fibrous bundles with smooth surfaces. The effect of different pretreatment times of ammonia on cellulose was further explored, and cellulose was characterized by Fourier transform infrared spectroscopy (FTIR), X-ray diffraction (XRD) and thermogravimetric (TG) analysis. The results showed that the N3h-CHB (NH_3_·H_2_O 50 °C 3 h, CH_3_COOH/H_2_O_2_ 70 °C 11 h, Bio-enzyme 50 °C 4 h) method was the best way to extract cellulose in this study. FTIR showed that most of the lignin and hemicellulose were removed. XRD showed that all the cellulose extracted in this study was type I cellulose. TG analysis showed that the cellulose was significantly more thermally stable, with a maximum degradation temperature of 338.9 °C, close to that of microcrystalline cellulose (MCC). This study provides a reference for the utilization of corn bract and offers a new technical route for cellulose extraction.

## 1. Introduction

With the increasing awareness of environmental protection among all people, the processing of agricultural wastes into versatile materials has received widespread attention in recent years. People are also gradually turning their attention from fossil fuel-based materials to renewable natural polymers such as cellulose [1]. Cellulose is available from a wide range of sources and has applications in several fields. For example, it has biomedical, pharmaceutical, composite, automotive, aerospace and packaging applications [2,3]. Cellulose is composed of D-glucopyranosyl groups linked by β-1,4-glycosidic bonds with the molecular formula (C_6_H_10_O_5_)_n_, and its structural formula is shown in Figure 1 [4,5].

Corn is widely grown around the world, and corn bract, as a by-product of corn, has many advantages. Corn bracts are soft, strong and slender, but they are often discarded directly or burned, which causes serious environmental pollution while wasting resources [6,7]. The main components of corn bract include cellulose, hemicellulose and lignin [8]. Among them, α-cellulose content is high at 36.28%, and lignin content is low at 14.78%, which means they can be used as a raw material source for cellulose extraction [9]. However, in lignocellulosic biomass, cellulose is often intertwined and interwoven with hemicellulose and lignin. Therefore, the key to cellulose extraction is to break the chemical or hydrogen bonds between lignin, hemicellulose and cellulose by pretreatment [10,11].

The reported cellulose extraction methods are mainly classified into physical, chemical and biological methods. The physical method is mainly used to separate cellulose from the raw material by destroying the structure of cellulose through external force, which generally has very limited extraction of cellulose and results in low purity. Higher-purity cellulose can be obtained by chemical treatment methods, but the process often also causes damage to cellulose and also has the problem of environmental pollution [12,13]. The biological method is a green and gentle pretreatment method that uses bio-enzyme specificity to degrade specific substances and retain cellulose components. However, due to the structural properties of biomass feedstock, it is resistant to bio-enzymes. Hemicellulose has been reported to inhibit enzyme activity [14], and lignin inhibits enzyme hydrolysis to varying degrees [15]. Therefore, to improve the effect of enzyme treatment, some pretreatment of the feedstock is required to increase the contact between the enzyme and the biomass feedstock. Ren et al. extracted purified MCC from sorghum using hydrochloric acid along with cellulase and xylanase [16]. Yılmaz successfully extracted corn husk cellulose by water retting, alkalinization and enzymatic methods at different concentrations and duration. The results showed that the enzymatic treatment and alkalization enhanced the thermal durability of the fibers [17].

Ammonia has effective lignin-solubilizing ability to remove acetyl groups and various glycolic acids from hemicellulose. In addition, ammonia has the effect of swelling the fiber structure. The swollen fiber structure becomes loose, which makes it easier for enzymes to approach the surface of its internal components and facilitates the enzymatic reaction. At the same time, the loose fiber can improve the permeability to distilled water during filtration, thus reducing the filtration time and energy consumption [18,19]. In recent years, a series of ammonia pretreatment methods have been studied and reported: ammonia fiber expansion (AFEX), ammonia recycle percolation (ARP) and soaking in aqueous ammonia (SAA) [20]. Zulkiple et al. used oil palm empty fruit bunch (OPEFB) as feedstock to evaluate the effects of different alkaline pretreatment methods on sugar production prior to enzymatic digestion. The results showed that ammonia pretreatment at 100 °C for 3 h using a pressurized chamber method significantly dispersed the lignin in the feedstock, thus increasing the delignification yield after hydrolysis [21]. Kim et al. investigated the pretreatment of pulp mill waste with ammonia and hydrogen peroxide. The results showed that about 30% of lignin and 50% of hemicellulose were removed from the feedstock by the ARP process [22]. Kim et al. used corn stover as raw material, and the results showed that the main effects of ammonia-soaking treatment were cellulose swelling and delignification [23].

At present, most of the studies on the ammonia-coordinated bio-enzymatic treatment of biomass feedstock are focused on the extraction of sugars or the preparation of bioethanol, but few are focused on the extraction of cellulose [24,25,26]. Furthermore, the existing studies often set high temperature, high pressure or soaking for a longer time in order to achieve the effect of lignin and hemicellulose removal. However, this may reduce the degree of polymerization and crystallinity of cellulose to some extent [27,28,29]. To avoid the occurrence of side reactions and reduce energy consumption and cost, a cellulose extraction method that can overcome these defects needs to be explored. In addition, more studies have been conducted on corn stover and corn cob, while corn bract has been less studied [25]. In this study, we investigated the feasibility of the ammonia-coordinated bio-enzymatic method for the extraction of cellulose from corn bract. Firstly, a milder ammonia pretreatment was set up, and then the corn bract was further treated with glacial acetic acid (CH_3_COOH)/hydrogen peroxide (H_2_O_2_) combined with bio-enzymes. Finally, the most suitable pretreatment time for ammonia was investigated in depth.

## 2. Materials and Methods

### 2.1. Materials and Reagents

Material: Corn bract was obtained locally (Kunming, Yunnan Province, China). It was ground to pass through a 40–60 mesh screen (250–420 µm) before cellulose extraction and chemical analysis.

Chemical reagents: Microcrystalline cellulose (MCC, conforming to Q/CYDZ 2320-2009) was obtained from Sinopharm Chemical Reagent Co., while ammonia (NH_3_·H_2_O, 25% *w/v*), hydrogen peroxide (H_2_O_2_, 30%) and glacial acetic acid (CH_3_COOH, 99.5%) were purchased from Yunnan Shuoyang Biological Company, Ltd., Shuoyang, China. Analytical purity-grade reagents were used in all experiments.

### 2.2. Extraction of Cellulose from Corn Bracts

The corn bract cellulose was extracted via two methods, based on the results of previous studies on the pretreatment of biomass materials, simplifying the single-factor experiments. The treatments with glacial acetic acid (CH_3_COOH)/hydrogen peroxide (H_2_O_2_) and bio-enzymes were fixed, and only the effect of ammonia (NH_3_·H_2_O) addition on the pretreatment of raw materials was considered. The specific process flow is shown in Figure 2.

#### 2.2.1. Method I: CHB (CH_3_COOH/H_2_O_2_/Bio-Enzyme)

Two-stage method: (1) C/H (CH3COOH/H2O2) treatment: 15 g of corn bract powder was placed in beakers and immersed in 300 mL of a 150:150 mixture of H2O2:CH3COOH at 70 °C until the sample turned white. The samples were washed with distilled water until the pH was 7. They were then oven-dried at 105 ± 2 °C to constant weight. (2) B (Bio-enzyme) treatment: 5 g of material from the previous step was weighed and immersed in 180 mL of a 60:60:60 mixture of 5% ligninase solution: 10% hemicellulase solution: 5% pectinase solution with stirring (900–1000 r/min) for 4 h at 50 °C. The samples were then neutralized by repeated washing with distilled water; they were then dried at 65 °C and stored in a desiccator for subsequent use. Ligninase (CAS: 80498-15-3), pectinase (CAS: 9032-75-1) and hemicellulase (CAS: 9025-56-3) were obtained from Aladdin Biotechnology Co., Ltd., (Shanghai, China), with enzyme activities of 500 U/g, 30,000 U/g and 5000 U/g, respectively.

#### 2.2.2. Method II: N-CHB (NH_3_·H_2_O-CH_3_COOH/H_2_O_2_/Bio-Enzyme)

Three-stage method: (1) N (NH3·H2O) treatment: 18 g of corn bract powder was immersed in 500 mL of 25% ammonia, and the container was closed airtight for 1 h and heated to a temperature of 50 °C. The samples were washed with distilled water until the pH was 7. The samples were then oven-dried at 105 ± 2 °C to constant weight. (2) C/H (CH3COOH/H2O2) treatment: 15 g of the material from the previous step was weighed in a beaker and immersed in 300 mL of a 150:150 mixture of H2O2:CH3COOH at 70 °C until the sample turned white. The samples were washed with distilled water until the pH was 7. They were then oven dried at 105 ± 2 °C to constant weight. (3) B (Bio-enzyme) treatment: 5 g of the material from the previous step was weighed and immersed in 180 mL of a 60:60:60 mixture of 5% ligninase solution: 10% hemicellulase solution: 5% pectinase solution with stirring (900–1000 r/min) for 4 h at 50 °C. The samples were then neutralized by repeated washing with distilled water; they were then dried at 65 °C and stored in a desiccator for subsequent use.

### 2.3. Properties of Cellulose

#### 2.3.1. Appearance Morphology Analysis

The results of each step in the cellulose extraction process were analyzed macroscopically to observe the changes in appearance, morphology and color.

#### 2.3.2. The Content of the Three Major Elements (Cellulose, Hemicellulose, Lignin)

Corn bract powder and cellulose samples were taken. Lignin content was determined according to the Chinese standard GB/T2677. 8-94 (“Determination of acid-insoluble lignin in paper raw materials”). Holo-cellulose content was determined according to the Chinese standard GB/T2677. 10-1995 (“Determination of holo-cellulose in paper raw material”). α-cellulose content was determined according to the Chinese standard GB/T744-1989 (“Determination of α-cellulose in pulp”). The hemicellulose content was calculated according to Equation (1), the experiment was repeated three times, and the results were averaged.
Hemicellulose content (%) = Holo-cellulose content (%) − α-cellulose content (%)(1)

#### 2.3.3. Fourier Transform Infrared Spectroscopy (FTIR) Analysis

The corn bract powder and cellulose were mixed with KBr, pressed into a pellet and analyzed on a Nicolet i50 FTIR Analyzer (Thermo Scientific, Waltham, MA, USA) with a scanning range of 500 to 4000 cm^−1^ and 64 scans.

#### 2.3.4. X-ray Diffraction (XRD) Analysis

The corn bract powder and cellulose were examined by X-ray diffractometry on a Rigaku Ultima IV X-ray diffractometer (Rigaku Corp, Tokyo, Japan) (XRD, Ulti) using a scanning angle from 10° to 40°, a step size of 0.026° (accelerating current = 30 mA and voltage = 40 kV) and Cu-Kα radiation of λ = 0.154 nm. The degree of crystallinity (DOC, %) was calculated according to the formula:(2)$$\mathrm{DOC}\left(\%\right)=\frac{I_{Max}-I_{Am}}{I_{Max}}\times 100\%\;$$

*I_Max_* is the maximum intensity of the main peak (about 22°), and *I_Am_* is the diffraction intensity of amorphous cellulose (about 15.8°).

#### 2.3.5. Thermogravimetric (TG) Analysis

The sample were analysis on a TGA92 thermogravimetric analyzer (KEP Technologies EMEA, Caluire, France). N_2_ was used as the shielding gas, and Al_2_O_3_ was used as the reference compound. The temperature was increased from 35 to 800 °C at a rate of 20 °C/min to generate a thermogravimetric curve.

#### 2.3.6. Sample Microstructure (SEM) Analysis

The corn bract powder and cellulose were placed on an aluminum grid and examined by field-emission scanning electron microscopy on a Nova Nano SEM 450 microscope (FEI, Hillsboro, OR, USA). At least five fields were examined for each sample.

## 3. Results and Discussion

### 3.1. Extraction Method

#### 3.1.1. Corn Bract Powder and Cellulose

The change in the appearance of the cellulose samples extracted by the N-CHB method with each stage of the treatment process can be observed in Figure 3. The corn bract was light brown in color, and the samples treated with ammonia in the first stage showed little color change compared to the raw material. This may be due to the insufficient degree of heating treatment when using ammonia at 50 °C for 1 h only. This stage only resulted in less removal of lignin from the raw material or other effects on the fiber. The color of the samples treated with CH_3_COOH/H_2_O_2_ in the second stage later became significantly lighter, which was caused by the bleaching effect of CH_3_COOH/H_2_O_2_ and the removal of most of the lignin from the raw material. After the third stage of bio-enzyme treatment, a pure and white sample of corn bract cellulose was obtained. This was due to the specificity of the bio-enzyme, which further degraded hemicellulose, lignin and other substances effectively, and the color became lighter. In conclusion, judging from the appearance, it is possible to extract corn bract cellulose by the N-CHB method.

#### 3.1.2. The Contents of the Three Major Elements

The main components of corn bract include cellulose, hemicellulose and lignin [8]. After determination, the results are presented in Table 1, where the cellulose, hemicellulose and lignin contents of corn bract powder were 36.28%, 39.43% and 14.78%, respectively. Compared with the corn bract powder, the cellulose content was significantly higher, and the hemicellulose and lignin contents were significantly lower for both methods. Among them, the N-CHB method reduced the hemicellulose content more significantly than in the cellulose extracted by the CHB method, and the lignin content was also reduced to some extent. Thus, it is clear that the synergistic effect of CH_3_COOH/H_2_O_2_ and bio-enzymes on the removal of lignin is obvious, but only a smaller portion of hemicellulose can be removed. The addition of ammonia pretreatment was made prior to the bio-enzymatic treatment. On the one hand, most of the hemicellulose in the corn bract was degraded. On the other hand, it made the lignin removal more complete. This is probably because ammonia degraded part of the lignin and hemicellulose. At the same time, the ammonia softened and swelled the corn bract. This facilitated the bio-enzymes approaching the interior of the fibers, thus further degrading the lignin, hemicellulose and pectin in the fibers.

#### 3.1.3. Fourier Transform Infrared Spectroscopy (FTIR)

To further analyze the effects of different pretreatment methods on the extraction of cellulose from corn bract. The infrared spectra of cellulose and MCC extracted from corn bract, the CHB method and the N-CHB method were compared and shown in Figure 4. The positions of the absorption peaks of curves a, b and c remained basically the same, which proved that the molecular structure of cellulose did not change too much before and after the extraction.

The curve is a stretching vibration peak of -OH near 3412 cm^−1^. Near 2901 cm^−1^ is the stretching vibration peak of C-H. The bending vibration peak of -OH is near 1649 cm^−1^, and this is the signal absorption peak of water molecules in cellulose [30]. The stretching vibration peak of C-O-C is present at 1053 cm^−1^. The characteristic absorption peak of β-glycosidic bond linkage in the cellulose molecule is at 895 cm^−1^, and these are the characteristic peaks of cellulose [31,32]. Curve a shows the characteristic peak of lignin corresponding to the aromatic ring carbon skeleton at 1506 cm^−1^ [33]. The peaks of curves b and c at 1506 cm^−1^ largely disappeared, indicating that both the CHB and N-CHB methods can largely remove lignin from corn bract. The peak at 1732 cm^−1^ is a C=O stretching vibration peak of COOH, which is characteristic of hemicellulose [34]. The peak of curve c at 1732 cm^−1^ is significantly weakened, and the peak of curve b there is also somewhat weakened. This result indicates that both the CHB and N-CHB methods can degrade hemicellulose, but the N-CHB method is more effective for hemicellulose degradation. Compared to curve b, curve c shows a significant increase in peak intensity at 1431 cm^−1^, which indicates a higher cellulose content extracted by the N-CHB method [35].

In summary, the purity of cellulose extracted by the N-CHB method was higher than that of cellulose extracted by the CHB method. This indicates that the synergistic effect of CH_3_COOH/H_2_O_2_ and bio-enzymes is obvious for the removal of lignin but difficult to establish for the removal of hemicellulose. The process of adding ammonia pretreatment before the CHB method treatment can effectively degrade hemicellulose and lignin, which has a significant effect on the extraction of pure cellulose. This is consistent with the results of the above analysis. In conclusion, the N-CHB method was determined as the extraction method for this study.

#### 3.1.4. Sample Microstructure

Figure 5 shows scanning electron microscopy images of corn bract and N-CHB cellulose samples magnified to different magnifications. It can be observed that significant changes occurred in the microscopic morphology of the corn bract powder after treatment. Cellulose, hemicellulose, lignin and other impurities in the raw material are interspersed, showing a disorganized lamellar shape and uneven surface.

After a series of treatments, the samples became bundled fibers. This may be because of the removal of lignin, hemicellulose and other substances during the treatment process, which exposed the internal structure of cellulose. However, a small amount of lumpy material was still present on the smooth surface of the cellulose. In addition, some folds and small pores were observed. These small pores facilitate the penetration and reaction of reagents inside the cellulose. Better functionalized modification and further utilization of cellulose were achieved. Next, in order to obtain the optimal extraction conditions for cellulose, different ammonia treatment times were set in the next step of the extraction study. The effect of ammonia treatment time on the removal rate of hemicellulose and lignin was investigated.

### 3.2. Exploration of the Pretreatment Time of Ammonia (NH_3_·H_2_O)

In this chapter, the optimal pretreatment time of ammonia in the N-CHB method was investigated, and the ammonia pretreatment time gradient was set as shown in Table 2.

#### 3.2.1. Fourier Transform Infrared Spectroscopy (FTIR)

Figure 6 shows the FTIR spectra of the four cellulose samples extracted by the N-CHB method. The positions of the absorption peaks in curves a, b, c and d remained basically the same. This result indicates that the cellulose did not change too much in terms of molecular structure.

The curve is a stretching vibration peak of -OH near 3412 cm^−1^. Near 2901 cm^−1^ is the stretching vibration peak of C-H. The bending vibration peak of -OH is near 1649 cm^−1^, and this is the signal absorption peak of water molecules in cellulose [30]. The stretching vibration peak of C-O-C is present at 1053 cm^−1^. The characteristic absorption peak of β-glycosidic bond linkage in the cellulose molecule is at 895 cm^−1^, and these are the characteristic peaks of cellulose [31,32]. The characteristic peak of lignin corresponding to the aromatic ring carbon skeleton is at 1506 cm^−1^ [33]. The peaks of the four curves at this point largely disappeared, indicating that the lignin was largely removed from all four cellulose samples after a series of chemical treatments. The peak at 1732 cm^−1^ is a C=O stretching vibration peak of COOH, which is characteristic of hemicellulose [34]. Compared with curve a, the peak of curve b at 1732 cm^−1^ is significantly weaker. The peaks of curves b, c and d at 1732 cm^−1^ showed a gradual weakening trend.

These changes clearly indicate that hemicellulose in corn bract was gradually degraded and removed with the increase of ammonia pretreatment time. Considering the cost issues regarding factors such as time and material, the N3h-CHB method can be considered the best cellulose extraction method for this study.

#### 3.2.2. X-ray Diffraction (XRD) Analysis

The cellulose samples were compared with MCC for XRD analysis. It can be observed in Figure 7 that the curves produce diffraction peaks at 2θ = 15.8°, 22.4° and 34.5°, which correspond to the (110), (002) and (040) crystallographic planes of cellulose type I, respectively [36]. This further indicates that cellulose was successfully extracted in this experiment.

The results of the crystallinity calculations are shown in Table 3, where the crystallinity of the cellulose samples tended to increase and then decrease with the increase of ammonia treatment time. Among the samples, when the ammonia pretreatment time increased from 1 h to 3 h, the percentage of crystallinity of cellulose increased accordingly. This is mainly due to the removal of a large amount of hemicellulose and lignin from the amorphous zone by the N-CHB approach as the ammonia treatment time increases. The cellulose structure was retained, which led to a relative increase in cellulose content [37]. Interestingly, the crystallinity showed a decreasing trend when the ammonia pretreatment time was increased to 5 h and 7 h. This may be due to the long pretreatment time, which resulted in the disruption of hydrogen bonds in the cellulose crystalline region. Compared to the degree of removal of hemicellulose and lignin from the amorphous region, the cellulose crystalline region was disrupted to a greater extent, thus causing a decrease in overall crystallinity. This is in agreement with the results of Kim et al., who used ammonia [38], and Nursyafiqah Zulkiple et al., who used alkaline solutions [21] for the pretreatment of biomass materials. Their results indicate that pretreatment has a rising/declining/dual effect on the crystallinity of cellulose.

In summary, the different pretreatment times of ammonia influenced the removal of hemicellulose and lignin. On the one hand, proper pretreatment time was helpful in avoiding side reactions that occur due to long processing times. On the other hand, it was also able to reduce the apparatus energy consumption and experimental cost. Therefore, the ammonia pretreatment of 3 h was determined as the most appropriate pretreatment time in this study. The N3h-CHB method was determined as the best cellulose extraction method in this study.

#### 3.2.3. Thermogravimetric (TG) Analysis

Cellulose, as a linear polymer linked by glucose units, easily forms crystalline regions within the molecular chain. Therefore, cellulose is thermally stable, with a thermal decomposition temperature between 275 °C and 350 °C [39]. As can be observed from Figure 8, the N3h-CHB samples showed better thermal stability compared to the corn bract. The degradation peaks (initial degradation temperature (T_on_) and maximum degradation temperature (T_max_)) of the N3h-CHB samples were elevated to higher temperatures, and the final residual mass was lower than that of the feedstock. This is due to a series of chemical treatments to remove hemicellulose, lignin and various impurities from the raw material. This increases the purity of the cellulose and thus the thermal stability of the cellulose [40,41]. The samples were closer to the MCC curve, indicating that the cellulose extracted by the N3h-CHB method has better thermal stability and higher purity.

## 4. Conclusions

The combined chemical and bio-enzymatic method used in this study combines the advantages of both chemical and bio-enzymatic methods. While the extraction process is green and gentle, it also achieves effective removal of lignin and hemicellulose. The ammonia pretreatment was performed before the bio-enzyme treatment, which softened the corn bract fibers and enhanced the efficiency of the synergistic action of the chemicals with the bio-enzyme. This had a positive effect on the extraction of cellulose.

The morphology of the raw material and cellulose was observed, the content of the three major elements was determined, and the cellulose was characterized by SEM, FTIR, XRD and TG analysis. The optimal treatment conditions for this study were determined as follows: NH_3_·H_2_O 50 °C 3 h, CH_3_COOH/H_2_O_2_ 70 °C 11 h and Bio-enzyme (ligninase/hemicellulase/pectinase) 50 °C 4 h.

After the treatment, no significant damage was caused to the cellulose of corn bract. The N3h-CHB method can be applied as a green and efficient method for cellulose extraction from other biomass resources. This not only reduces the cost of raw materials, but also has important significance for environmental protection and resource utilization.

## Figures and Tables

**Figure 1 polymers-15-00206-f001:**
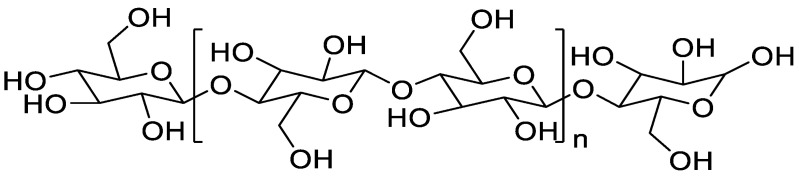
Molecular structure formula of cellulose (Note: n is the degree of polymerization).

**Figure 2 polymers-15-00206-f002:**
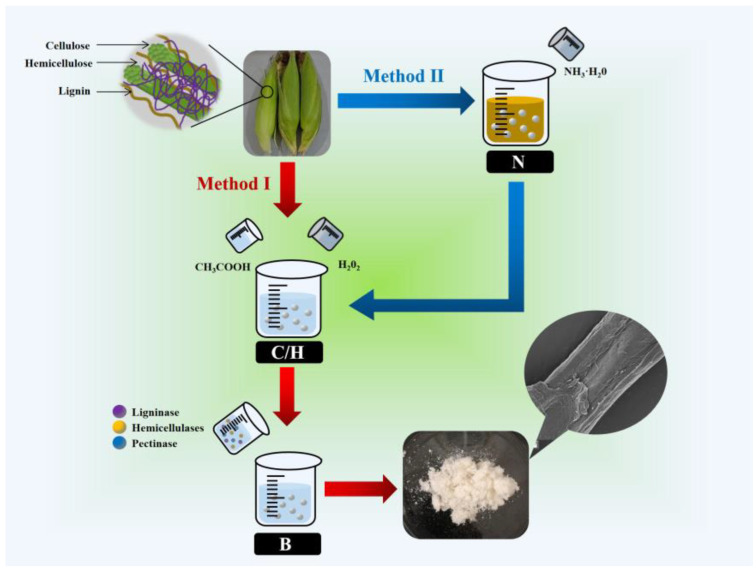
Experimental flow chart (N: NH_3_·H_2_O; C/H: CH_3_COOH/H_2_O_2_; B: Bio-enzyme).

**Figure 3 polymers-15-00206-f003:**
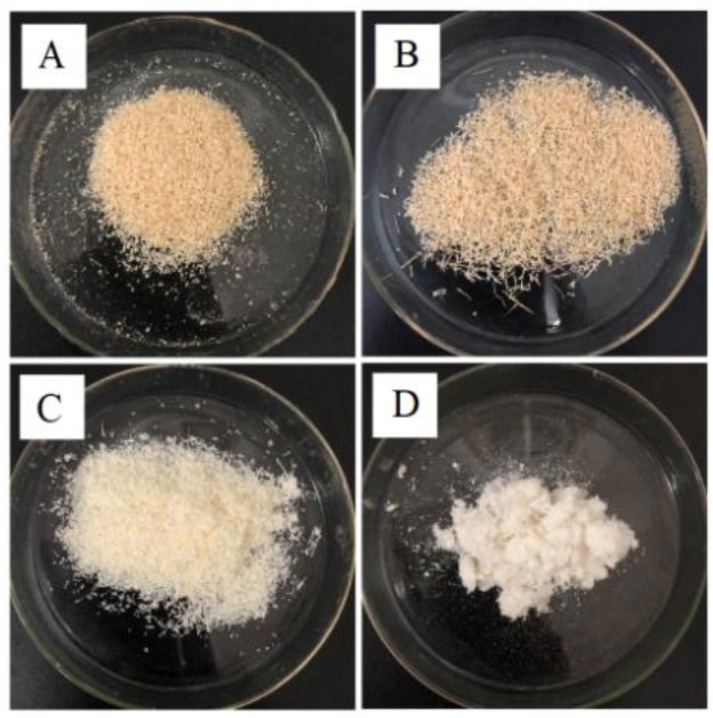
Appearance morphology of each step in the N-CHB cellulose extraction process: (**A**) Corn bract powder; (**B**) Sample treated with NH_3_·H_2_O; (**C**) Sample further treated with CH_3_COOH/H_2_O_2_; (**D**) N-CHB cellulose.

**Figure 4 polymers-15-00206-f004:**
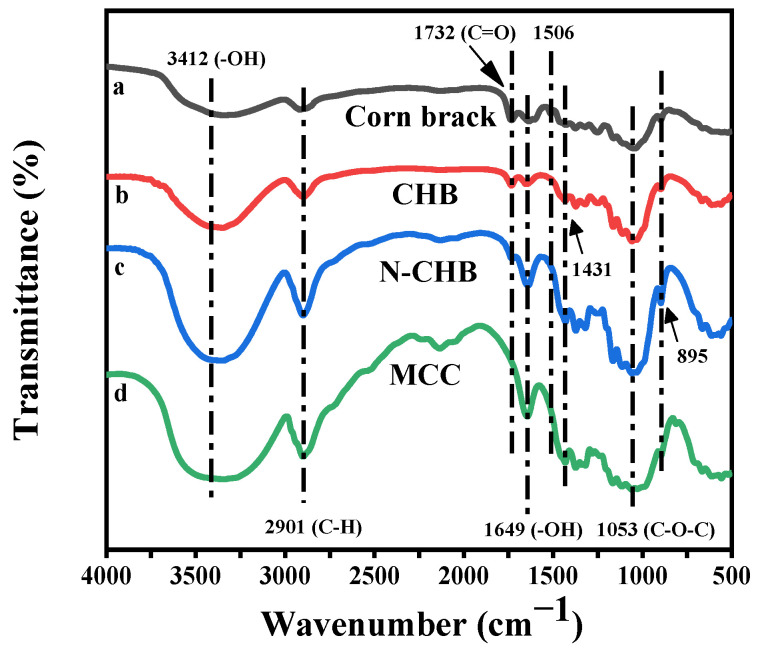
Infrared spectra: (a) Corn bract; (b) CHB cellulose; (c) N-CHB cellulose; (d) MCC.

**Figure 5 polymers-15-00206-f005:**
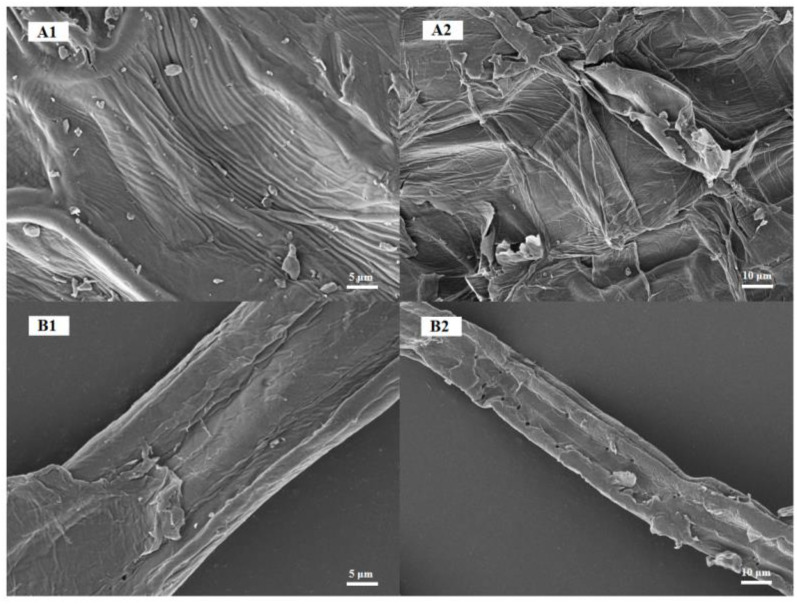
SEM of Corn bract and N-CHB cellulose: (**A1**) Corn bract powder, 5 μm; (**A2**) Corn bract powder, 10 μm; (**B1**) N-CHB cellulose, 5 μm; (**B2**) N-CHB cellulose, 10 μm.

**Figure 6 polymers-15-00206-f006:**
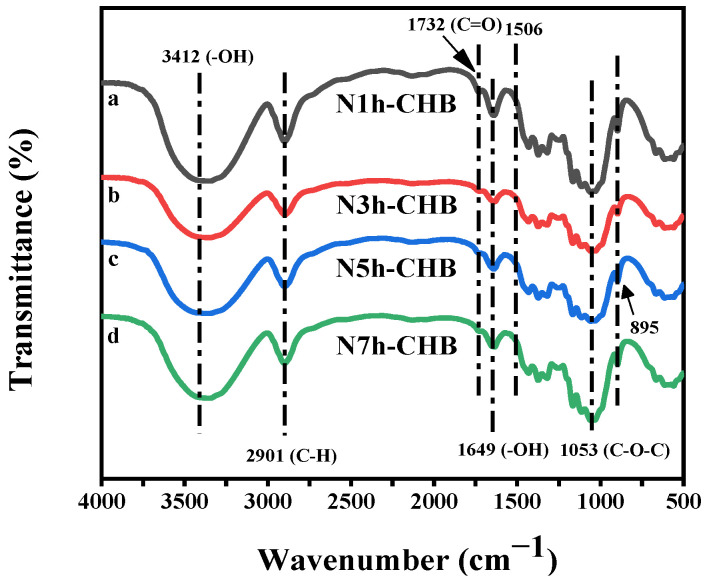
Infrared spectra: (a) N1h-CHB cellulose; (b) N3h-CHB cellulose; (c) N5h-CHB cellulose; (d) N7h-CHB cellulose.

**Figure 7 polymers-15-00206-f007:**
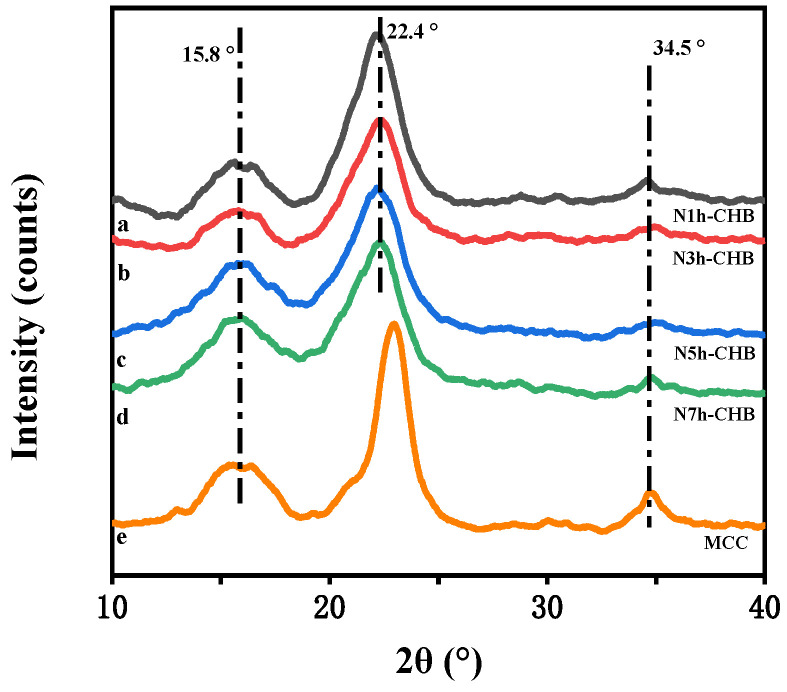
XRD spectra: (a) N1h-CHB cellulose; (b) N3h-CHB cellulose; (c) N5h-CHB cellulose; (d) N7h-CHB cellulose; (e) MCC.

**Figure 8 polymers-15-00206-f008:**
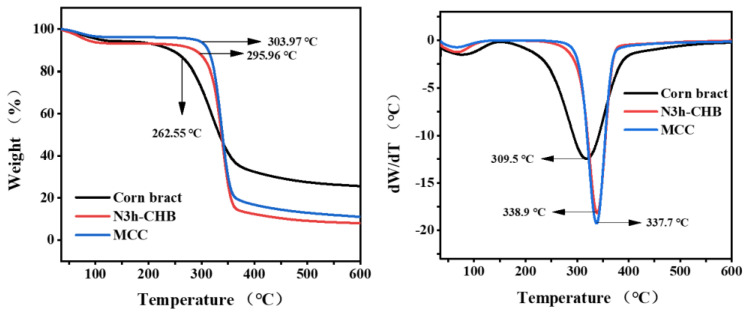
TG/DTG curves of corn bract, N3h-CHB cellulose and MCC.

**Table 1 polymers-15-00206-t001:** Content of the three major elements in corn bract powder and cellulose samples *.

Name	Cellulose (%)	Hemicellulose (%)	Lignin (%)	Hemicellulose Removal Rate (%)	Lignin Removal Rate (%)
Corn bract powder	36.28 (0.41)	39.43 (0.19)	14.78 (0.37)	-	-
CHB cellulose	52.16 (0.34)	35.16 (0.42)	3.98 (0.93)	10.83 (0.24)	73.07 (0.37)
N-CHB cellulose	75.66 (0.52)	12.95 (0.63)	2.12 (0.29)	67.16 (0.31)	85.66 (0.50)

*: Numbers in parentheses are standard deviations.

**Table 2 polymers-15-00206-t002:** Solution preparation and pretreatment method.

Test No.	Test Name	NH_3_·H_2_O Treatment Time/h	NH_3_·H_2_O Treatment Temperature/°C
1	N1h-CHB	1	50
2	N3h-CHB	3	50
3	N5h-CHB	5	50
4	N7h-CHB	7	50

**Table 3 polymers-15-00206-t003:** The crystallinity of four cellulose samples and MCC.

Test Name	*I* * _Max_ *	*I* * _Am_ *	DOC (%)
N1h-CHB	1334.465	337.691	74.695
N3h-CHB	1042.954	201.698	80.661
N5h-CHB	1358.574	650.962	52.085
N7h-CHB	1360.979	685.597	49.625
MCC	2546.406	717.176	71.836

## Data Availability

The data presented in this study are available on request from the corresponding author.

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
