# Peer review of "Extraction of Corn Bract Cellulose by the Ammonia-Coordinated Bio-Enzymatic Method"

_polymers, 2022, doi:10.3390/polym15010206_

Round 1

Reviewer 1 Report (Previous Reviewer 2)

The manuscript can be accepted.

Author Response

Title: Extraction of corn bract cellulose by the ammonia-coordinated bio-enzymatic method

Manuscript ID: polymers-2000224

Response to Reviewer Comments

Comments and Suggestions for Authors: The manuscript can be accepted.

Thank you very much. Some grammatical errors and unclear sentences in the manuscript were modified. In addition, we have revised parts of the article based on the check report.

Reviewer 2 Report (Previous Reviewer 1)

The presented study on extraction of corn bract cellulose by ammonia- coordinated bio-enzymatic method. The manuscript had been greatly improvedThe manuscript should be minor revised before publishing on Polymers considering the following remarks:

1. In section 2, the author should present the properties of the used MCC.

2.In table 2, it might be more important that it should show the results of the relationship of NH3 ·H2O treatment time and the removal rate of the hemicellulose and lignin. What is the result?

3. Did the concentration of NH3 ·H2O has the effect on the crystallinity of cellulose and the hemicellulose and lignin removal rate.

4. The tense of the manuscript should be improved.

Author Response

Title: Extraction of corn bract cellulose by the ammonia-coordinated bio-enzymatic method

Manuscript ID: polymers-2000224

Response to Reviewer Comments

The presented study on extraction of corn bract cellulose by ammonia- coordinated bio-enzymatic method. The manuscript had been greatly improved. The manuscript should be minor revised before publishing on Polymers considering the following remarks:

Thank you very much for your email. We appreciate the useful comments and suggestions of reviewers, and have revised the manuscript accordingly.

Comments: 1. In section 2, the author should present the properties of the used MCC.

Response: Thank you for your advice. We had changed this part as followed:

  1. Materials and Methods

2.1. Materials and Reagents

Material: corn bract were obtained locally (Kunming, Yunnan Province, China). It was ground to pass through a 40-60 mesh screen (250-420 µm) before cellulose extrac-tion and chemical analysis.

Chemical reagents: Microcrystalline cellulose (MCC, conforming to Q/CYDZ 2320-2009) was obtained from Sinopharm Chemical Reagent Co., Ammonia (NH3·H2O, 25% w/v), Hydrogen peroxide (H2O2, 30%), Glacial acetic acid (CH3COOH, 99.5%) were purchased from Yunnan Shuoyang Biological Company, Ltd, Shuoyang, China. Analytical purity-grade reagents were used in all experiments.

Comments: 2. In table 2, it might be more important that it should show the results of the relationship of NH3·H2O treatment time and the removal rate of the hemicellulose and lignin. What is the result?

Response: Thank you for your advice, that help us to improve the quality of our manuscript.

In Table 2, four experimental schemes with different ammonia pretreatment times in this study are listed. A detailed analysis of the relationship between NH3·H2O treatment time and the degree of hemicellulose and lignin removal is presented in sections 3.2.1, 3.2.2 and 3.2.3.

Among them, the FTIR results showed that the peaks at 1506 (the characteristic peak of lignin) of the curves of the four cellulose samples basically disappeared. It indicates that the lignin has been largely removed from all of them. The smoother the peak at 1732 (the characteristic peak of hemicellulose) as the treatment time increases. It shows that the longer the treatment time of NH3·H2O, the cleaner the removal of hemicellulose. The XRD results showed that the crystallinity of the cellulose samples tended to increase and then decrease with the increase of NH3·H2O treatment time. To avoid the occurrence of side reactions caused by prolonged treatment and to reduce energy consumption and cost, NH3·H2O treatment for 3 h was determined as the better cellulose extraction method in this study. The TG results showed that the N3h-CHB samples obtained from NH3·H2O treatment for 3 h were thermally stable and of high purity.

In addition, regarding the relationship between ammonia treatment time and the degree of hemicellulose and lignin removal. We have made some modifications in the article to highlight the results of the relationship.

3.2.2. X-ray diffraction (XRD) analysis

In summary, the different pretreatment times of ammonia influenced the removal of hemicellulose and lignin. On the one hand, to avoid side reactions that occur due to long processing times. On the other hand, it was also to reduce the apparatus energy consumption and experimental cost. Therefore, the ammonia pretreatment of 3 h was determined as the most appropriate pretreatment time in this study. The N3h-CHB method was determined as the better cellulose extraction method in this study.

Comments: 3. Did the concentration of NH3 ·H2O has the effect on the crystallinity of cellulose and the hemicellulose and lignin removal rate.

Response: Thank you for your comments and we will follow your guidance.

The concentration of ammonia may have some effect on the extraction of cellulose. In future studies, different concentrations of ammonia may be considered to be explored and compared with the current study. Based on other researchers in the literature who have used ammonia at a concentration of about 25%, this study only explored the treatment time of ammonia in depth and fixed the concentration at 25%.

Birgit Kammet et al. used the AFEX process to study the effect of ammonia (25% w/v) on glucose conversion during enzymatic hydrolysis. The results show that the process can achieve more than 90% conversion of glucose and avoid carbohydrate degradation products. It can be used as a simple and inexpensive method for some agricultural structures [1]. KIM et al. pretreated corn stover with Soaking in aqueous ammonia (SAA) using 29.5% ammonia. The results showed that SAA treatment removed 55-74% of lignin and some hemicellulose [2]. KIM et al. investigated the effect of ammonia reaction temperature and concentration on SAA. The results showed that an increase in ammonia concentration from 15% to 30% had little effect on lignin removal and for a slight increase in enzymatic digestibility of the samples [3]. Seiichi Inoue et al. chose cellulose and ammonia as the simplest carbon and nitrogen sources, respectively, and investigated the effect of N/C on the liquefaction process of nitrogen in biomass. In the liquefaction process, ammonia acted as both reactant and alkaline catalyst. Where the ammonia concentration was 25% [4].

Comments: 4. The tense of the manuscript should be improved.

Response: Some grammatical errors and unclear sentences in the manuscript were modified. In addition, we have revised parts of the article based on the check report.

Thank you again for your precious comments and suggestions.

References

[1] Kamm, B., Leiß, S., Schönicke, P., & Bierbaum, M. Biorefining of lignocellulosic feedstock by a modified ammonia fiber expansion pretreatment and enzymatic hydrolysis for production of fermentable sugars. ChemSusChem2017. 48-52.

[2] Kim, T.H.; Lee, Y.Y. Pretreatment of corn stover by soaking in aqueous ammonia. Appl. Biochem. Biotechnol. 2005, 121, 1119-1131.

[3] Kim, Tea Hyun, and Y. Y. Lee. Pretreatment of corn stover by soaking in aqueous ammonia at moderate temperatures. Applied Biochemistry and Biotecnology. Humana Press, 2007. 81-92.

[4] Inoue, S., Okigawa, K., Minowa, T., & Ogi, T. Liquefaction of ammonia and cellulose: Effect of nitrogen/carbon ratio in the feedstock. Biomass and Bioenergy, 1999. 377-383.

This manuscript is a resubmission of an earlier submission. The following is a list of the peer review reports and author responses from that submission.

Round 1

Reviewer 1 Report

The presented study on extraction of corn bract cellulose by ammonia- coordinated bio-enzymatic method. This study found that NH3·H2O pretreatment was attributed to the extraction of cellulose. However, the overall concept is not new or novel. Therefore, the manuscript is not suitable for publishing on Polymers considering the following remarks:

1. its obvious that the pretreatment of biomass by acid or base is attributed to the extraction of cellulose. so the innovation of the manuscript was low.

2. In lines 86-96, the relevant literatures should be cited.

3. In line 125, what is E treatmet.

4. In line 352-353, it could not be found from SEM that hemicellulose and lignin and other impurities were effectively removed from the cellulose.

5. the conclusion was too complicated and should be rewrote.

6. The English and the format of the manuscript was terrible.

Reviewer 2 Report

The novelty of the proposed work is very less. The authors fail to explain the role (a mechanism) of ammonia in the extraction and separation of cellulose. My other comments are as follows:

1. Several good reports exist in the same field, so authors must highlight the proposed strategy's advancement over others.

2. What is the effect of ammonia concentrations on the activity of bio-enzymes? conduct a separate experiment and explain. Since it is very crucial in this study.

3. In Figure 6: Provide XRD plot for pure cellulose to compare the crystallinity of obtained cellulose.

4. section 2.2.2 It is not clearly mentioning that whether the reaction vessel was covered or open, If it was not refluxed then ammonia gets evaporated at 50 oC over a period of reaction time.